# Data quality in centenarian research: The proxy-centenarian relationship and item nonresponse in the SWISS100 study

**Daniele Zaccaria**[1]*, **Justine Falciola**[2], **Barbara Masotti**[1,3], **Armin von Gunten**[4], **François Herrmann**[2], **Daniela S. Jopp**[3,5], **Stefano Cavalli**[1]

**1** Department of Business Economics, Centre of Competence on Ageing, Health and Social Care, University of Applied Sciences and Arts of Southern Switzerland, Manno, Switzerland, **2** Faculty of Medicine, Department of Rehabilitation and Geriatrics, Geneva University Hospitals and University of Geneva, Geneva, Switzerland, **3** LIVES Swiss Centre of Expertise in Life Course Research, University of Lausanne and University of Geneva, Geneva, Switzerland, **4** Department of Psychiatry, Service of Old Age Psychiatry, Lausanne University Hospital and University of Lausanne, Lausanne, Switzerland, **5** Institute of Psychology, University of Lausanne, Lausanne, Switzerland

* daniele.zaccaria@supsi.ch

**Data Availability Statement:** All data files used in this study are available from the SWISSUbase repository, managed by FORS (The Swiss Centre of Expertise in the Social Sciences). The dataset is

## Abstract

### Background

In recent years, there has been a growing interest in the investigation of very old individuals. However, various challenges arise when collecting data from this age group. Given potential health and cognitive impairments and the difficulty of retrieving accurate self-reported data, involving individuals knowledgeable of the target person as proxy respondents are an invaluable solution. The primary goal of this study is to explore the association between proxy-centenarian relationship and item nonresponse, a pivotal data quality measure.

### Data and methods

We used data from the phone study conducted within SWISS100, a study on centenarians in Switzerland, focusing on a subsample of 94 centenarians and 75 proxy respondents. We compared characteristics of centenarians who provided self-reports with those interviewed through proxy respondents using descriptive analyses, and Firth's regression models to investigate the association of different types of proxies (children, other relatives, and non-relatives) with item nonresponse.

### Results

We noted differences between centenarians participating themselves in interviews and those represented by proxies, observing higher proportions of men and private home residents in the self-report group. There was a low prevalence of item nonresponse among proxy respondents. Proxy type significantly impacted item nonresponse, particularly with non-relatives showing notably higher unanswered question rates. A robust association between non-relatives as proxies and increased item nonresponse persisted even when controlling for potential confounding factors.

titled "SWISS100 Phone Study" (ref. no. 2707) and is part of the larger study, "SWISS100 – The First Nationwide Study on Centenarians in Switzerland" (ref. no. 20896). The dataset can be accessed via the following DOI: https://doi.org/10.48573/zm2a-b654.

**Funding:** The SWISS100 Study is financed by the Swiss National Science Foundation (SNSF) within the Sinergia Project "Swiss Centenarian Study: Vulnerability and resilience at age 100" (CRSII5_186239/1).

**Competing interests:** The authors have declared that no competing interests exist.

## Discussion and conclusions

This study enhances the understanding of data quality, particularly item nonresponse, and highlights the impact of the proxy-centenarian relationship when studying the oldest-old. The findings stress the importance of carefully choosing proxy respondents, preferably children, to improve data quality and inclusivity for those individuals hard to reach or to interview. Future research should investigate various data quality indicators and rely on larger samples to enhance the representation and methodological rigour in the study of the oldest-old.

## Introduction

In recent years, the study of individuals reaching very advanced ages, particularly centenarians, has gained substantial traction in gerontology and epidemiology, driven by an increasing interest in potential implications for healthcare and society at large [1–3]. This growing interest has led to an increase in research aimed at understanding the unique characteristics, experiences, and health profiles of individuals who have reached 100 years of age [4–6]. Yet, scholars in this field are confronted with many challenges when it comes to collecting accurate and comprehensive data from this growing demographic group.

The challenges are manifold: poor health status and frailty, potential sensorial and cognitive impairments, and increased fatigue while participating in an interview make self-reported data collection a difficult task. Furthermore, previous studies have shown that in Western countries at least one in five of those aged 80 or over lives in an institution such as retirement or nursing homes and that prevalence rises to at least 30% among people living with dementia [7,8]. In addition, the access to institutions of external individuals, such as researchers or interviewers, can be restricted by gatekeepers or caregivers that want to protect the centenarians [9]. For all these reasons the very old belong to the so called "hard-to-reach" population, more precisely, they represent a typical example of the sub-category of "hard-to-interview" population due to their specific characteristics [10].

The inclusion of proxy respondents, individuals who provide information on behalf of centenarians, offers a pragmatic solution to overcome these obstacles. Indeed, proxy respondents can serve as conduits of valuable information when the centenarians themselves are unable to participate fully in interviews.

While involving proxy respondents can indeed facilitate data collection, it introduces a critical concern: data quality. Previous evidence in the field of survey methodology has identified a wide range of indicators of data quality [11]. In this paper, we concentrate on a particularly critical aspect of centenarian research, namely the potential influence of using proxy respondents on data quality, with a specific focus on item nonresponse (INR). INR refers to the phenomenon where respondents omit or fail to answer specific survey questions [12].

Through analysing data collected within the Swiss Centenarian Study (SWISS100), the first nationwide study on centenarians in Switzerland, we seek to shed light on the proxy respondents' involvement and its implications for data quality. More specifically, we aim to assess to what extent different types of proxy respondents affect INR, as a paramount indicator. In the following sections, we will review the literature on data quality and data collection when investigating oldest-old people, generally defined as individuals aged 85 and older, as outlined by Kydd et al [13]. Then, we will outline our study's methodology, present our findings, and discuss their implications for research practice. By undertaking this investigation, we aim to

respond to the call for more research addressing the impact of proxy respondents on survey validity [14], while also providing guidance for researchers who navigate the complex terrain of surveying the oldest-old.

## Background

### Item nonresponse as an indicator of data quality

Data of old and oldest populations may be affected by various sources of errors of representation and measurement [15]. Even a well-designed study will produce biased results if the collected information are correlated with the probability of some groups or individuals to participate in the data collection or not. The use of proxy interviews allows researchers to gather information about those who are less likely to be recruited and interviewed, such as old and very old individuals living in institutional settings and those with more health issues [8,16]. However, to provide valid and reliable answers proxy respondents have to successfully search for the most relevant information, retrieve information from memory, compute judgments on specific situations. If respondents do not run through the question-answer-process as thoroughly as possible, they may not provide answers that most precisely reflect their judgments. This "satisficing behaviour" [17] has been frequently used to explain response effects and bias. Specifically, strong satisficing occurs when respondents completely skip the tasks of retrieval or judgement, leading to INR, one of the main indicators of measurement error that negatively affects quality of data [18,19]. INR can be considered a second-level nonresponse, in contrast to unit nonresponse, which occurs when no data at all is obtained for a respondent [20].

The survey methodology literature usually distinguishes three different types of INR, based on the reasons for the lack of information: "don't know", 'refusal', and 'no answer'. In the case of 'don't know' a respondent is willing but unable to respond to a question, maybe because of memory problems when asked about past behaviour or the lack of an opinion on a particular issue. Even though the 'don't know' option may not always be interpreted as INR since it may reflect the absence of a real opinion [21], it is mainly interpreted as a form of a satisficing behaviour occurring when respondents have low ability or low motivation to exert the cognitive efforts necessary to answer a question or when questions are particularly difficult [17,22]. When the respondent has the relevant information but is not willing to answer (e.g., because of the sensitive nature of the information) this is a typical example of 'refusal'. The third type of INR–'no answer'–can be considered a residual category. It is usually less intentional since it may contain for instance missing values due to the fact that the interviewer or the respondent did not correctly follow the skip patterns of a questionnaire and thereby missed one or several questions [19]. Several scholars in their studies on old and very old populations have operationalized the INR indicator by combining at least two of these three categories, most of the times 'don't know' and 'refusal' options [23–26].

Regardless of its operationalization, INR is one of the major concerns when collecting data since it decreases the available sample size for analyses and can lead to the risk of biased results if the missingness does not occur at random [20].

### The use of proxy respondents to collect data about the oldest-old

Proxy respondents, often family members, caregivers, or designated surrogates, step in to provide information on behalf of the primary subject [27]. This practice becomes especially prevalent when dealing with very old individuals, such as centenarians, whose potential health, sensorial, and cognitive limitations may negatively affect their ability to participate fully in

surveys [28–30]. While the use of proxy respondents for the old and oldest-old is rooted in practicality, its potential implications for data quality are not well understood [31].

Most of the studies investigating the methodological implications of the use of proxies as data sources have focused so far on the analysis of the agreement between proxy and target respondents and the extent to which the characteristics of these proxy respondents may affect data quality [32]. The critical dimension that received more attention is the nature of the relationship between target participants and their proxies, although with mixed results. Recruiting a spouse as a proxy respondent has been associated with more reliable data in some studies [33,34], while others reported better results when the answers have been given by older people's offspring or other family members [35,36] because partners of very old adults can be themselves old and frail [37]. Furthermore, in a study of old and oldest-old adults' health, professional proxy provided better data quality compared to spouses and other family proxies [38]. In addition, in a study on functional abilities and continence there was a tendency for more distant relationship to produce better agreement between older women and their proxy respondents [26]. Other studies reported that higher agreement was associated with being male or older than 60 years [39], living with or visiting more often the target participant [29,40], or less subjective burden due to caregivers duties [41].

Surprisingly, despite the key role of INR as an indicator of bias and measurement error, few scholars focused on this topic when analysing the quality of data collected through the proxies of old and oldest-old adults. Hansen and colleagues [23], in their study about quality of life among German oldest-old, reported that INR was low, i.e., about 4%, and mainly due to non-valid answers given by proxy respondents. Similarly, in a study on older women aged from 66 to 100 years living in England, INR was also low–i.e., less than 3%–for all questions asked to proxy respondents, who were mainly children of target respondents [26]. In their discussion of data collection procedures and challenges with the oldest-old, Rodgers and Herzog [42] stressed the importance of the relationship between target population and proxy respondents in determining the level of INR, especially regarding subjective information. The type of proxy respondent emerged as the primary factor influencing the completeness of responses also in the study by Pickle and colleagues [43]. Specifically, when considering sociodemographic characteristics of both target and proxy respondents, it was observed that siblings were more effective in responding to questions related to the subject's immediate family or events from early life, while spouses and offspring were better suited to describe events from the subject's adult life.

All the above results refer to target populations composed of older adults, while there is limited evidence on the very old, such as centenarians, even though research on this population often uses proxy ratings instead of self-reports [44–47]. MacDonald and colleagues [48] conducted a study examining the correspondence between a sample of American centenarians and their proxies regarding mental health ratings. They found no significant differences between mean values provided by the different raters. Another study involving 53 couples of German centenarians and their proxies reported a high degree of similarity in some health ratings [49]. However, neither study delved into the investigation of INR and its association with proxy respondents' characteristics.

In sum, our analysis of the literature reveals a notable gap in the examination of the characteristics of proxy respondents, particularly in the context of centenarian research, and their potential impact on INR. More specifically, despite the recognized importance of the proxy-target relationship, the exploration of how different types of relationships may affect INR when collecting data on the very old remains relatively unexplored across various disciplines, such as survey methodology, epidemiology, and gerontology. The target respondent-proxy relationship, characterized by its depth, familiarity, and emotional bonds, has the potential to

significantly influence the completeness of responses. In order to advance our understanding in this domain, our study placed a central focus on this critical variable. First, we described the characteristics of centenarians who participated in the phone survey within the SWISS100 study, to evaluate to what extent those who answered directly differed from those who were replaced by a proxy respondent. This allowed us to assess whether the exclusion of centenarians who could not participate directly in the data collection would lead to a potential bias of representativeness in the sample, particularly the under-representation of certain categories, such as the frailest individuals or those living in nursing homes. Then, we focused on the association between INR and the type of proxy respondents, accounting for some potential confounding factors, to assess the independent role of the relationship between centenarians and their proxies in influencing the probability of INR. Through this analysis, we could shed light on which proxies would be best to engage in future studies on the very old to collect more complete and higher quality data even in the presence of hard-to-reach and hard-to-interview groups.

## Data and methods

### The study sample

We based this study on data from SWISS100 study. It is a study aimed at identifying specific characteristics, challenges, and needs of Swiss centenarians, adopting a multidisciplinary perspective, since it investigates biological, health, psychiatric, psychological, and social aspects of the life at this age. SWISS100 involves centenarians as the target population and, for each of them, a proxy respondent, for instance, a family member or a person who has a solid knowledge of the target participant, such as friends or formal caregivers. The study started in 2020 but, due to the Covid-19 outbreak, the data collection, conducted through face-to-face interviews with centenarians and proxies, as well as a paper-and-pencil self-completed questionnaire, was delayed. Waiting for the situation to improve from December 15, 2020, to June 2022, we conducted an exploratory cross-sectional study with telephone interviews. The study protocol was approved by the Swiss Ethics Committees on research involving humans (ID 2020–02063) and was conducted in accordance with the Declaration of Helsinki and with the principles and procedures for integrity in scientific research involving human beings.

The study sample of the phone survey included both centenarians, i.e., older adults aged 100 years or older, living in Switzerland, and proxy respondents. Centenarians were contacted according to a pre-established random order within each Swiss canton, based on a list provided by the Federal Statistical Office. Prior to initiating telephone contact, an invitation letter was sent to all potential centenarian participants in the study. Then, centenarians were contacted via telephone, and those who did not speak one of the Swiss main national languages, namely French, German, or Italian, were excluded. Additionally, individuals who were unable to participate directly in the interview due to severe cognitive impairment, and lacked an available proxy respondent, were also excluded. Specifically, if the telephone contact with the centenarian was not feasible due to issues such as hearing or cognitive problems, trained interviewers inquired whether the person on the phone could answer questions about the centenarian. If not, they requested an alternative contact who might be able to provide information about the centenarian, without specifying the relationship, given the exploratory nature of the telephone study. In the case of institutionalized centenarians unable to participate directly in the interview, the interviewers required from the staff the contact information of a close family member. If no family members were available, it was allowed to recruit as a proxy respondent either a non-kin person or a staff member, both of whom should have a good knowledge of the centenarian. Recruited proxies provided information not only about

centenarians who were unable to participate directly in the interview but also about themselves. An information sheet containing all the study characteristics and the rights of each participant was sent after scheduling the date of the interview. Each phone interview, lasting about 60 minutes, began with the acquisition of verbal informed consent and was recorded after the participant's approval. At the conclusion of each interview, interviewers assessed the respondents' cooperation and the need for additional clarifications. In total, our data collection encompassed 169 centenarians residing in 22 cantons, out of 26 Swiss cantons. Specifically, our sample included 94 centenarians and 75 proxy respondents who stepped in for those centenarians unable to participate directly.

## Variables

Our dependent variable pertains to INR for each proxy respondent during the interview. To address the issue of a small sample size, we aggregated non-valid responses, including 'don't know' and 'refusal,' when calculating the total number of such responses for each proxy respondent, as previously done in survey methodology research [50]. These responses were collected across a set of 14 questions about the centenarian, covering various aspects such as sociodemographic factors (i.e., age, gender, marital status, education, and living situation), self-reported health, a short version of the Geriatric Depression Scale (5-GDS) consisting in 5 items with yes/no answers [51], and 3 items from the Valuation of Life (VOL) scale (i.e., whether the centenarian has religious beliefs, has a strong will to live, and experiences the life as meaningful), measured on a 5-point scale ranging from 0 (not at all) to 4 (very much) [52]. First, we calculated the proportion of INR. Then, owing to the highly skewed distribution of INR, we converted the total INR count for each respondent into a binary variable format. In this format, '0' indicated the absence of any INR, while '1' represented the presence of at least one non-valid answer in the responses, consistent with the approach employed by previous scholars in their studies on INR. [25,53].

The independent variable pertains to the relationship between the centenarian and the proxy respondent. The original variable provided by SWISS100 data distinguished between sons, daughters, stepchildren, grandchildren, nieces, nephews, spouses, other relatives, friends, acquaintances, formal caregivers, nurses, and nursing home staff members. Given the limited sample size, we reconfigured these categories into a new three-category variable: 'children' (son, daughter, stepchild), 'other relatives' (spouse, grandchild, nephew, niece), and 'non-relatives' (friend, acquaintance, formal caregiver, nurse, nursing home staff member). The rationale for this choice extends beyond the scarcity within specific categories (e.g., only one spouse among the recruited proxies) to encompass a deliberate focus on the offspring of centenarians. Research suggests that centenarians often rely on their children for primary support in their daily lives [54]. This could make children an accessible and willing group for recruitment as proxy respondents. As for the other proxies, we have distinguished between those related to the centenarian and those who aren't. As mentioned earlier, this separation potentially leads to variations in the quality and accuracy of the collected data.

The control variables belong to four domains: characteristics of proxy respondents, characteristics of centenarians, proxy respondents' behaviour during the interview, and data collection features. *Characteristics of proxy respondents* include gender (0 = female, 1 = male), age at interview, marital status (0 = unmarried, divorced, 1 = married, 2 = widow, widower), education (0 = less than high school, 1 = high school or more), and self-reported health (SRH) (0 = poor, fair, 1 = good, 2 = very good, excellent). *Characteristics of centenarians* are age at interview, gender, living conditions (0 = home, 1 = nursing home), and health reported by

their proxies (0 = poor, fair, 1 = good, 2 = very good, excellent). *Proxy respondent's behaviour during the interview*, an essential factor influencing satisficing and respondent ability to provide accurate responses [17,55], was assessed by interviewers at the interview's conclusion. Specifically, cooperation, originally rated on a 5-point Likert scale ('very good' to 'very poor'), was recoded as follows to account for valid answers: 0 = very good, 1 = good or fair. Also request for clarifications, evaluated using a 6-item scale ('never' to 'always'), was recoded as follows: 0 = never, 1 = at least once or twice. *Data collection features* include year of interview (0 = 2020–2021, 1 = 2022) as a proxy of the variations in the stringency measures implemented to contain the COVID-19 pandemic, and the language-speaking region where the data was collected (0 = French, 1 = German, 2 = Italian).

## Data analysis

To start, we analysed the characteristics of our centenarian sample, looking at age, sex, marital status, education, living situation, health, and depression. Descriptive statistics–i.e., percentages for categorical variables and means with standard deviations for continuous variables–were used to summarize the data. We compared centenarians who provided self-reports to those for whom proxy reports were obtained using appropriate statistical tests. For categorical variables, Pearson's chi-squared or Fisher's exact test was used, depending on the sample size. For continuous variables, the Mann-Whitney U test, a non-parametric alternative to the t-test, was employed to account for non-normal distributions.

Next, we assessed the prevalence of INR and its association with the proxy-centenarian relationship using the Kruskal-Wallis test (a non-parametric alternative to the analysis of variance) for the proportion of unanswered questions, and Fisher's exact test for the binary INR variable. Subsequently, to evaluate the independent association between the relationship of centenarians and proxy respondents with INR, we employed a Firth's logistic regression model. This method was selected due to its suitability for the specific characteristics of our sample, which includes a small sample size and potential issues of data separation. Firth's logistic regression is an adaptation of standard logistic regression designed to handle situations where traditional methods might struggle, particularly in the presence of small samples or rare events. In such cases, standard logistic regression can yield biased parameter estimates or even fail to converge, especially when a predictor variable perfectly separates the outcome categories–i.e., when the predictor perfectly predicts the outcome. To mitigate these issues, Firth's logistic regression applies a penalized likelihood estimation technique. This adjustment reduces the bias in the parameter estimates by penalizing the likelihood, making the estimates more robust in the context of small or imbalanced samples. This is particularly important in our study, where the sample size may limit the reliability of traditional logistic regression results. Moreover, the use of Firth's method ensures that the model can accommodate a variety of independent variables, whether they are continuous, categorical, or a combination of both, without the risk of inflated or biased estimates. This leads to more reliable and interpretable results, which are less likely to be unduly influenced by outliers or the rarity of certain events within the sample [56,57].

We initially ran an unadjusted model, followed by four models adjusting for different sets of control variables: a) characteristics of proxy respondents, b) characteristics of centenarians, c) proxy respondents' behaviour during the interview, and d) data collection features. Additionally, we conducted checks for multicollinearity among covariates in each model by testing variance inflation factors and tolerance criteria. These checks revealed no issues of collinearity (see S1 Table for complete values). We performed all the analysis using Stata 16.0 [58].

## Results

### Differences in centenarians' characteristics based on self-vs proxy reports

We analysed differences in characteristics of centenarians who participated directly in the interview and those for whom we obtained proxy respondent reports (Table 1). There were significant differences in the living situation, $\chi^2$ (1, N = 169) = 8.5, p = .003, and there were trends in the distribution of gender across the two groups, $\chi^2$ (1, N = 169) = 5.7, p = .017. More specifically, in the centenarian self-reports group there were significantly more men (31.9%) and individuals living in private homes (45.7%) compared to those for whom we collected information through proxy respondents (16.0% and 24.0%, respectively). No significant differences were found regarding other characteristics. Although we found no significant differences for subjective health, $\chi^2$(2, N = 165) = 0.652, p = 0.652, and depression scores, t(163) = 1.5, p = 0.091,, their numeric values suggested a trend towards poorer evaluations of health and higher depression levels when the information was provided by a proxy respondent replacing the centenarian (SRH: 45.4% vs. 42.2%, 5-GDS: M = 1.8, SD = 1.6 vs. M = 1.3, SD = 1.4).

### Characteristics of proxy respondents by relationship with centenarians

Table 2 presents key characteristics of proxy respondents categorized by their relationship with the centenarians. The largest group comprised the children of centenarians (n = 51), followed by non-relatives (n = 14) and other relatives (n = 10). It is noteworthy that, among centenarians for whom the proxy was not a child, only one of them had no living children. Unfortunately, it is not possible to determine why the children of those centenarians were not recruited in place of the other relatives or non-relatives. The Kruskal-Wallis test indicated a statistically significant difference in age among the three groups, $\chi^2$ (2, N = 74) = 6.2, p = .045,

**Table 1. Characteristics of centenarians by source of information.**

|  | Total | Self-report | Proxy-report |  |
|---|---|---|---|---|
|  | % (n) | % (n) | % (n) | p |
| *Age*, mean (SD) | 101.8 (1.7) | 101.6 (1.5) | 102.0 (1.9) | 0.304 |
| *Gender* |  |  |  |  |
| Men | 24.8 (42) | 31.9 (30) | 16.0 (12) | 0.017 |
| Women | 75.2 (127) | 68.1 (64) | 84.0 (63) |  |
| *Marital status* |  |  |  |  |
| Without partner | 14.8 (25) | 14.9 (14) | 14.7 (11) | 0.893 |
| Married | 9.5 (16) | 8.5 (8) | 10.7 (8) |  |
| Widow/er | 75.7 (128) | 76.6 (72) | 74.7 (56) |  |
| *Education* |  |  |  |  |
| Less than high school | 74.5 (108) | 72.8 (59) | 76.6 (49) | 0.610 |
| High school or more | 25.5 (37) | 27.2 (22) | 23.4 (15) |  |
| *Living situation* |  |  |  |  |
| Home | 36.1 (61) | 45.7 (43) | 24.0 (18) | 0.003 |
| Nursing home | 63.9 (108) | 54.3 (51) | 76.0 (57) |  |
| *Self-reported health* |  |  |  |  |
| Poor, fair | 45.4 (75) | 42.2 (38) | 49.3 (37) | 0.652 |
| Good | 37.6 (62) | 38.9 (35) | 36.0 (27) |  |
| Very good, excellent | 17.0 (28) | 18.9 (17) | 14.7 (11) |  |
| *5-GDS*, mean (SD) | 1.5 (1.5) | 1.3 (1.4) | 1.8 (1.6) | 0.081 |

**Table 2. Characteristics of proxy respondents by relationship with centenarians.**

| | Total (N = 75) | Children (n = 51) | Other relatives (n = 10) | Non-relatives (n = 14) | p |
|---|---|---|---|---|---|
| | % (n) | % (n) | % (n) | % (n) | |
| *Age*, mean (SD) | 67.3 (11.3) | 69.8 (7.7) | 69.8 (10.8) | 56.9 (16.5) | 0.045 |
| *Sex* | | | | | |
| Male | 38.7 (29) | 35.3 (18) | 30.0 (3) | 57.1 (8) | 0.284 |
| Female | 61.3 (46) | 64.7 (33) | 70.0 (7) | 42.9 (6) | |
| *Marital status* | | | | | |
| Without partner | 37.3 (25) | 29.2 (14) | 60.0 (6) | 55.6 (5) | 0.070 |
| Married | 56.7 (38) | 66.7 (32) | 30.0 (3) | 33.3 (3) | |
| Widow/er | 6.0 (4) | 4.2 (2) | 10.0 (1) | 11.1 (1) | |
| *Education* | | | | | |
| Less than high school | 33.8 (24) | 38.8 (19) | 30.0 (3) | 16.7 (2) | 0.336 |
| High school or more | 66.2 (47) | 61.2 (30) | 70.0 (7) | 83.3 (10) | |
| *Self-rated health* | | | | | |
| Poor, fair | 13.2 (9) | 14.6 (7) | 20.0 (2) | 0.0 (0) | 0.712 |
| Good | 42.7 (29) | 43.7 (21) | 30.0 (3) | 44.4 (4) | |
| Very good, excellent | 44.1 (30) | 41.7 (20) | 50.0 (5) | 55.6 (5) | |

with children of centenarians (M = 69.8, SD = 7.7) and other relatives (M = 69.8, SD = 10.8) being older on average compared to non-relatives (M = 56.9, SD = 16.5). No statistically significant differences were observed in sex, marital status, education, or subjective health of the proxies. It could however be the case that the children who did not participate in the study had by themselves health issues, given that they are also of advanced age. We could not test this assumption due to lack of information on these non-participants.

### Item non-response rates across proxy types

With regards to the total amount of INR across the 14 questions we considered, the overall rate of unanswered questions among all proxy respondents in the sample was 4% (Table 3). The Kruskal-Wallis test, $\chi^2$ (2, N = 75) = 8.5, p = .014, showed that there was a statistically significant difference in the average rate of INR between the three types of proxy respondents. Specifically, children of centenarians (M = 3.4, SD 8.1) and other relatives (M = 2.9, SD = 4.9) had lower rate of INR than non-relatives (M = 7.1, SD = 6.9)–see also S1 Fig. S2 Table provides the absolute and relative distribution of INR for each variable considered. While the INR among the socio-demographic questions concerns only information related to education, most of the INR is due to subjective questions, specifically the VOL items asking whether the centenarian has religious beliefs, a strong will to live, and experiences meaning in life, and even more the GDS depressive symptom items. Table 3 also presents the derived INR variable

**Table 3. Item nonresponse by relationship proxy-centenarian.**

| Item nonresponse | Children | Other relatives | Non-relatives | Total | p |
|---|---|---|---|---|---|
| Mean (SD) | 3.4 (8.1) | 2.9 (4.9) | 7.1 (6.9) | 4.0 (7.6) | 0.014 |
| Derived variable | % (n) | % (n) | % (n) | % (n) | |
| *Complete answers* | 80.4 (41) | 70.0 (7) | 35.7 (5) | 70.7 (53) | 0.004 |
| *At least one nonresponse* | 19.6 (10) | 30.0 (3) | 64.3 (9) | 29.3 (22) | |
| Total | 51 | 10 | 14 | 75 | |

and shows that the share of proxy respondents who did not answer all questions, i.e., having at least one INR, was 29.3%. Furthermore, there was a notable pattern regarding INR among proxy respondents and their relationship with centenarians: there was a significant association between the type of proxy respondent and INR, as revealed by the Fisher's exact test (p = .004). Specifically, non-relatives serving as proxies for centenarians showed a much higher likelihood of INR, with 64.3% of cases having at least one unanswered item. This percentage is considerably higher compared to children (19.6%) and other relatives (30.0%) of the centenarians.

## The association between item non-response and the type of proxy respondents

The results of the Firth's logistic models (Table 4) demonstrated a significant pattern in the association between INR and the type of proxy respondents, even when controlling for potential confounding factors. Although odds ratios generally showed a positive association across all models, when comparing the odds ratios for other relatives engaged as proxy respondents to the reference category (i.e. being children of the centenarian), they did not reach statistical significance without or with control variables (no control [M0]: OR = 1.84, 95% CI [0.44–7.77]; proxy characteristics as controls [M1]: OR = 2.94, 95% CI [0.55–15.70]; centenarian's characteristics as controls [M2]: OR = 1.97, 95% CI [0.43–8.91]; proxy behaviour during interview as control [M3]: OR = 2.26, 95% CI [0.50–10.18]; data collection features [M4]: OR = 1.98, 95% CI [0.43–9.15]). In contrast, despite the wide confidence intervals, the presence of non-relatives as proxy respondent consistently exhibited a statistically significant positive association with INR across models (M0: OR = 6.8, 95% CI [1.96–23.82]; M1: OR = 12.62, 95% CI [1.86–85.47]; M2 (OR = 6.63, 95% CI [1.70–25.85]; M3: OR = 6.10, 95% CI [1.22–30.14]; M4: OR = 6.25, 95% CI [1.67–23.35]). Thus, even controlling for potential confounding factors, the association between being a non-relative proxy and INR remained generally consistent with the unadjusted model suggesting a robust and persistent association between this type of proxy respondent and INR. Complete Firth's regression models are available in S3

**Table 4. Firth's logistic regression models showing the association of item nonresponse with proxy-centenarian relationship.**

|  | M0 | M1 | M2 | M3 | M4 |
|---|---|---|---|---|---|
|  | OR [95% CI] | OR [95% CI] | OR [95% CI] | OR [95% CI] | OR [95% CI] |
| Children [a] |  |  |  |  |  |
| Other relatives | 1.84 [0.48–8.70] | 2.94 [0.55–15.70] | 1.97 [0.43–8.91] | 2.26 [0.50–10.18] | 1.98 [0.43–9.15] |
| Non-relatives | 7.54** [2.13–26.73] | 12.62** [1.85–85.47] | 6.63* [1.70–25.85] | 6.10* [1.22–30.14] | 6.25** [1.67–23.35] |
| Control variables |  |  |  |  |  |
| a) Characteristics of proxy respondents |  | ✓ |  |  |  |
| b) Characteristics of centenarians |  |  | ✓ |  |  |
| c) Proxy respondents' behaviour during the interview |  |  |  | ✓ |  |
| d) Data collection features |  |  |  |  | ✓ |
| N | 75 | 67 | 75 | 66 | 75 |

*Notes*: OR = Odds ratio

[a] = reference category.

Significance level

** = p < .010

* = p < .050.

Table and S2 Fig. provides graphical representation of the predicted probabilities of INR across the five models.

## Discussion

Our study is situated within the ongoing debate about data quality when investigating hard-to-reach and hard-to-interview populations, with centenarians representing a particularly challenging group in this context. We contribute to the existing literature by shedding light on the understudied association between INR as an essential indicator of data quality and the characteristics of proxy respondents.

First, our results indicate a significant difference in living situations between centenarians interviewed directly and those interviewed through proxy respondents, with centenarians living private setting were more likely to provide self-reports, whereas centenarians in nursing home settings were more likely to be replaced by a proxy respondent. This finding may indeed suggest a potential sample representativeness bias if we had excluded those unable to participate directly in the study, a concern previously highlighted by scholars who emphasized the importance of including oldest-old individuals residing in long-term care facilities such as nursing homes [8,9]. Relying on proxy respondents enables the inclusion of centenarians who might have been too frail or cognitively too limited to participate directly in the interviews. While not statistically significant, our results showed a trend in line with expectations that centenarians who would have been excluded without the opportunity to participate through a proxy respondent tended to exhibit poorer physical and mental health. It is important to note, however, that previous research has highlighted the risk of overestimating negative health conditions when relying on proxy respondents for older individuals [34,59]. Nevertheless, the approach employed for the SWISS100 data collection undeniably guaranteed a more inclusive and comprehensive representation of the Swiss centenarian population.

Considering our results on quality of data given the involvement of proxy respondent, two distinct but interrelated aspects warrant attention and further discussion. Firstly, it's noteworthy that our study reveals a generally low prevalence of INR, which is consistent with findings from other previous research [23,26]. Thus, including proxy respondents is associated with a general increase in important information, and missingness is quite limited when choosing this approach. This may be because due to the study design, we only took into account a subset of questions and that many of these concerned socio-economic conditions, for which answering is relatively easy even for proxies who know little about the centenarian (e.g., gender or place of living). However, if we focus on the rate of respondents who did not provide complete answers, we have almost a third of them with at least one INR. Plausible explanations for this phenomenon could be linked to the nature of the unanswered questions and the interview mode. The SWISS100 questions addressed to proxy respondents that mainly contributed to INR asked for subjective topics, more specifically the evaluation of psychological states, beliefs or attitudes towards life (i.e., depression, religious beliefs, will to life and meaning in life). These topics may have presented a greater challenge for respondents, resulting in a higher occurrence of unanswered questions. This aligns with the notion that respondents are more likely to skip or avoid questions that touch upon subjective or sensitive matters, a phenomenon that has been observed in various survey research involving the general population [60]. Moreover, as indicated by previous findings in the field of survey methodology, telephone interviews may place a greater cognitive burden on respondents than face-to-face interviews, particularly when dealing with subjective topics [22,53].

Secondly, our findings strongly emphasize the significance of the relationship between proxy respondents and the target population, centenarians in our case. In this regard, our

results align with scholars who underscore the importance of this factor as a primary determinant of the data quality collected through proxies for old and oldest-old individuals. [42]. We observed that, unlike non-relatives, children acting as proxy respondents consistently provided more comprehensive responses, irrespective of individual characteristics, target population conditions, and study design features. Additionally, although not statistically significant, our results suggest a tendency according to which other relatives may also offer more complete data compared to non-relatives. This implies that they could be a preferred type of proxy respondents, especially in scenarios where, for instance, a centenarian has no children or they are unavailable for recruitment. This result concerning INR contributes to the broader debate on data quality and the use of proxy respondents in the study of old and oldest-old individuals. Specifically, it aligns with that of scholars who, while focusing more on other indicators of data quality such as proxy reliability or agreement with the target respondents, emphasize that being a close relative, especially a child, often leads to the provision of more accurate and complete data [35,36]. In addition, even though previous studies primarily focusing on populations younger than those involved in SWISS100 have shown that age can lead to an increased risk of INR [19,37], our result supports the argument that the effect of age is more negligible when considering the relationship with the target respondent [43]. This is because children, despite being older adults themselves, were more likely to provide complete data compared to younger non-relative proxy respondents.

## Limitations

While this study offers valuable insights, there are several limitations that warrant consideration. First, the study's small sample size, consisting of 75 proxy respondents, may limit the generalizability of the findings to a broader population. This small sample also led to wider confidence intervals, reducing the precision of our estimates and necessitating cautious interpretation of the results [61–63]. Additionally, due to sample size constraints, a fully adjusted model with all control variables could not be estimated, which may have left certain confounding factors unaddressed. Second, our approach to measuring INR involved combining 'don't know' and 'refusal' responses into a single category. While this was necessary to manage the small sample size, it may have obscured important distinctions between these types of non-responses. Separating these categories in future studies could provide more nuanced insights into the factors contributing to INR [19,64].

Third, the use of telephone interviews, while necessary due to the circumstances of the COVID-19 pandemic, may have placed a greater cognitive burden on respondents, particularly when dealing with subjective questions. This mode of data collection could have contributed to the observed INR, especially for questions requiring assessments of psychological and emotional states.

Addressing these limitations in future research will be essential for enhancing the robustness and generalizability of findings related to data quality in studies involving hard-to-reach and hard-to-interview populations like centenarians.

## Conclusion

This study is among the few to investigate the methodological implications of using proxy respondents in centenarian research. Our findings underscore that the type of proxy respondent can significantly affect data quality, particularly INR, in surveys of the oldest-old, such as centenarians. We demonstrate that proxies with close familial ties, such as adult children, generally provide more accurate and complete data, which is crucial for achieving representative samples of this hard-to-reach population. Although this may not be true for all types of

questions, and the quality of the resulting answers may vary, this strategy can help alleviate biases that might arise due to the exclusion of individuals with physical or cognitive limitations or those residing in nursing homes. However, it is essential to conduct further research to comprehensively assess the impact of proxy responses on data quality, ideally in the context of larger samples, in order to gain a more nuanced insights on the underlying factors. This research should also consider various indicators of data quality, such as rounding, acquiescence, and straightlining, as this will further contribute to expanding our understanding and reducing uncertainty in the results.

For future research involving proxy respondents of the very old, we recommend addressing potential data quality issues, from the study's design phase to the analysis stage, to contribute to reinforcing the robustness of the findings. We also encourage researchers to continue exploring innovative data collection methods aimed at improving the representation and accuracy of information on the oldest-old population.

## Supporting information

**S1 Table. Variance inflation factors and tolerance criteria.**
(PDF)

**S2 Table. Distribution of unanswered questions across variables.**
(PDF)

**S3 Table. Firth's logistic regression models showing the association of item nonresponse with proxy-centenarian relationship, controlled for potential confounders.**
(PDF)

**S1 Fig. Average INR rate by type of proxy respondent.**
(TIF)

**S2 Fig. Mean predicted probability of INR by type of proxy in models M0 to M4.**
(TIF)

## Author Contributions

**Conceptualization:** Daniele Zaccaria, Barbara Masotti, Stefano Cavalli.

**Data curation:** Justine Falciola.

**Formal analysis:** Daniele Zaccaria.

**Funding acquisition:** Armin von Gunten, François Herrmann, Daniela S. Jopp, Stefano Cavalli.

**Writing – original draft:** Daniele Zaccaria.

**Writing – review & editing:** Daniele Zaccaria, Justine Falciola, Barbara Masotti, Armin von Gunten, François Herrmann, Daniela S. Jopp, Stefano Cavalli.

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
