## [Decision Letter · Decision Letter 0]

17 Jul 2024

PONE-D-24-19896Data quality in centenarian research: The proxy-centenarian relationship and item nonresponse in the SWISS100 studyPLOS ONE

Dear Dr. Zaccaria,

Thank you for submitting your manuscript to PLOS ONE. After careful consideration, we feel that it has merit but does not fully meet PLOS ONE’s publication criteria as it currently stands. Therefore, we invite you to submit a revised version of the manuscript that addresses the points raised during the review process.

We look forward to receiving your revised manuscript.

Kind regards,

Satabdi Mitra, M.D. (Community Medicine), DNB (SPM), MNAMS 

Academic Editor

PLOS ONE

Journal Requirements:

3. Thank you for stating the following financial disclosure: "The SWISS100 Study is financed by the Swiss National Science Foundation (SNSF) within the Sinergia Project “Swiss Centenarian Study: Vulnerability and resilience at age 100”  (CRSII5_186239/1)."

4. In the online submission form, you indicated that "Data cannot be shared publicly because of [privacy issues]. Data are available upon request from Prof. Daniela Jopp (Daniela.Jopp@unil.ch) for researchers who meet the criteria for access to confidential data."

Reviewers' comments:

Reviewer's Responses to Questions

**Comments to the Author**

1. Is the manuscript technically sound, and do the data support the conclusions?

Reviewer #1: Partly

Reviewer #2: Yes

2. Has the statistical analysis been performed appropriately and rigorously? 

Reviewer #1: Yes

Reviewer #2: Yes

3. Have the authors made all data underlying the findings in their manuscript fully available?

Reviewer #1: No

Reviewer #2: Yes

4. Is the manuscript presented in an intelligible fashion and written in standard English?

Reviewer #1: Yes

Reviewer #2: Yes

5. Review Comments to the Author

Reviewer #1: It is a good work from the authors. But the aim of the study is not clear and correlating with the final conclusion of this study. The methods part needs to be elaborated with clear description of how the analysis of the study was done. What is oldest-old classification? The authors need to add more graphical representation of the results for clear understanding to the readers. The limitation of the study needs to be highlighted. The conclusion need to have the main finding of the study from the results.

Reviewer #2: In the research paper titled "Data Quality in Centenarian Research: The Proxy-Centenarian Relationship and Item Nonresponse in the SWISS100 Study," the author explores the relationship between proxies and centenarians. The article adeptly examines mental health among the elderly.

The paper is professionally written,but there are some formatting issues such as inconsistent text borders and lack of justification.

Additionally, the referencing style varies, noticeable in the discrepancy with the 6th reference compared to others.

Overall, I recommend acceptance pending minor revisions.

6. PLOS authors have the option to publish the peer review history of their article (what does this mean?). If published, this will include your full peer review and any attached files.

Reviewer #1: **Yes: **SARIKA GOPALAKRISHNAN

Reviewer #2: No

---

## [Author Response · Author response to Decision Letter 0]

24 Sep 2024

PONE-D-24-19896

Data quality in centenarian research: The proxy-centenarian relationship and item nonresponse in the SWISS100 study

Response to reviewers

Thank you for the opportunity to revise our manuscript “Data quality in centenarian research: The proxy-centenarian relationship and item nonresponse in the SWISS100 study”. We appreciate the feedback provided by the reviewers and the academic editor. We have carefully considered each comment and have made the necessary revisions to improve the clarity, rigor, and overall quality of our paper. Below, we provide a detailed response to each point raised, outlining the changes made and, where appropriate, offering additional clarification.

 Journal Requirements:

1. Please ensure that your manuscript meets PLOS ONE's style requirements, including those for file naming

We have reviewed the PLOS ONE style templates provided at the links mentioned and have adjusted our manuscript files accordingly to ensure they meet the journal’s formatting and file naming requirements. The manuscript, as submitted, now adheres to all PLOS ONE style guidelines.

We have reviewed the grant information to ensure consistency between the ‘Funding Information’ and ‘Financial Disclosure’ sections. The correct grant statement to be mentioned is: "The SWISS100 Study is financed by the Swiss National Science Foundation (SNSF) within the Sinergia Project “Swiss Centenarian Study: Vulnerability and resilience at age 100” (CRSII5_186239/1)".

3. Please state what role the funders took in the study.

The correct statement about the role of the funders in our study, as per your guidance, is as follows: "The funders had no role in study design, data collection and analysis, decision to publish, or preparation of the manuscript." This statement should be included in the revised manuscript, and we confirm that this accurately reflects the funders' involvement.

4. In the online submission form, you indicated that "Data cannot be shared publicly because of [privacy issues]. Data are available upon request from Prof. Daniela Jopp (Daniela.Jopp@unil.ch) for researchers who meet the criteria for access to confidential data." All PLOS journals now require all data underlying the findings described in their manuscript to be freely available to other researchers.

We have addressed the data availability requirement by depositing the dataset, "SWISS100 Phone Study" (ref. no. 2707), on the public repository SWISSUbase, which is managed by FORS (The Swiss Centre of Expertise in the Social Sciences). This dataset is part of the study "SWISS100 – The First Nationwide Study on Centenarians in Switzerland" (ref. no. 20896). The dataset has already been assigned a DOI: https://doi.org/10.48573/zm2a-b654 and it has the following citation:

Jopp, D., Cavalli, S., Von Gunten, A., Herrmann, F., Röcke, C., Falciola, J., Hoffman, A., Masotti, B., Uittenhove, K., & Zaccaria, D. (2024). SWISS100 Phone Study (Version 1.0) [Data set]. FORS. https://doi.org/10.48573/zm2a-b654.

This procedure ensures that all data underlying the findings are freely accessible to other researchers, fully complying with PLOS ONE's data availability policy. However, due to internal procedures at FORS, the web link to access the dataset has not yet been activated but is expected to become available in the coming days, most likely by the end of September. While this timeline extends beyond the submission deadline for our revised manuscript, we will promptly inform the journal as soon as the link is live to facilitate the publication process. In the meantime, we await the editor's guidance regarding how to proceed. 

5. Please include captions for your Supporting Information files at the end of your manuscript, and update any in-text citations to match accordingly. 

We have updated the manuscript to include captions for all Supporting Information files as requested. The captions provide clear descriptions of the content of each file. Additionally, we have revised all in-text citations to ensure they correspond to the updated captions according to PLOS ONE’s Supporting Information guidelines.

Here are the updated captions added to the end of the manuscript:

• S1 Table. Variance inflation factors and tolerance criteria. 

• S2 Table. Distribution of unanswered questions across variables. 

• S3 Table. Firth's logistic regression models showing the association of item nonresponse with proxy-centenarian relationship, controlled for potential confounders.

• S1 Fig. Average INR rate by type of proxy respondent. 

• S2 Fig. Mean predicted probability of INR by type of proxy in models M0 to M4. 

All references to Supporting Information files in the text have been updated to reflect these changes.

6. Please review your reference list to ensure that it is complete and correct.

We have thoroughly reviewed the reference list to ensure that it is both complete and accurate.

Reviewers’ comments:

Reviewer 1 Comment 1: The aim of the study is not clear and correlating with the final conclusion of this study.

We have revised the introduction to more explicitly state the primary objective of the study, which is to explore the association between item nonresponse and the type of proxy respondents in a centenarian population. We have ensured that the conclusion now directly correlates with this aim by summarizing the key findings related to this objective, emphasizing the implications for data quality in research involving the oldest-old populations. The revised text can be found in the Introduction and Conclusion sections.

Reviewer 1 Comment 2: The methods part needs to be elaborated with clear description of how the analysis of the study was done.

We have revised the "Data and Methods" section to include a more thorough explanation of the analytical strategies employed, specifically focusing on the rationale for choosing Firth’s logistic regression, and how we handled the small sample size and potential data separation issues. Additionally, we have provided a step-by-step description of the procedures for conducting our analysis, both descriptives and multivariate. These changes should clarify how the analysis was conducted and the rationale behind them.

Reviewer 1 Comment 3: What is oldest-old classification?

We recognize that the term "oldest-old" may require further clarification. The term "oldest-old", despite a lack of consensus among scholars, refers in our study to individuals who are 85 years and older, a subgroup of the older population that also includes centenarians (those aged 100 years and above). We have mentioned this threshold in the Introduction section, and the appropriate reference (i.e., Kydd et al. 2020), to ensure clarity for all readers.

Reviewer 1 Comment 4: The authors need to add more graphical representation of the results for clear understanding to the readers.

We have addressed this suggestion by providing the readers with additional graphical representations as supporting information to the main manuscript. Specifically, we included a chart of average rate of item nonresponse by type of proxy respondents (S1 Fig.) and a bar chart illustrating the mean predicted probability of INR by type of proxy for each estimated model, which visually complements the statistical findings (S2 Fig.). These graphs should enhance the reader's ability to interpret and understand the key results of our study.

Reviewer 1 Comment 5: The limitation of the study needs to be highlighted.

We have expanded the discussion of the study’s limitations in the revised manuscript. The limitations section now more explicitly addresses the small sample size, the potential biases introduced by relying on proxy respondents, and the constraints imposed by the operationalization of INR. We also discuss the implications of these limitations for the generalizability and interpretation of our findings. This expanded discussion can be found in a new specific paragraph titled “Limitations” in the latter part of the Discussion section.

Reviewer 1 Comment 6: The conclusion needs to have the main finding of the study from the results.

We have revised the Conclusion section to ensure that it clearly summarizes the main findings of the study, directly linking them to the study’s aim. The conclusion now succinctly reiterates the key results, particularly the relationship between the type of proxy respondent and the likelihood of INR. This ensures that the conclusion is aligned with both the study’s objectives and its results.

Reviewer 2 Comment 1: The paper is professionally written, but there are some formatting issues such as inconsistent text borders and lack of justification. Additionally, the referencing style varies, noticeable in the discrepancy with the 6th reference compared to others.

We have thoroughly revised the document to address the formatting issues identified. Specifically, we have corrected the inconsistent text borders, ensured that the entire manuscript is fully justified, and standardized the referencing style throughout, including the 6th reference, to ensure consistency with all other citations.

These revisions have been made to align the manuscript with the journal's formatting standards, and we are confident that it now meets the required professional and stylistic criteria.

---

## [Editor Report · Decision Letter 1]

25 Sep 2024

Data quality in centenarian research: The proxy-centenarian relationship and item nonresponse in the SWISS100 study

PONE-D-24-19896R1

Dear Dr. Daniele Zaccaria,

We’re pleased to inform you that your manuscript has been judged scientifically suitable for publication and will be formally accepted for publication once it meets all outstanding technical requirements.

Kind regards,

Satabdi Mitra, M.D(Community Medicine )

Academic Editor

PLOS ONE
---

## [Editor Report · Acceptance letter]

17 Jan 2025

PONE-D-24-19896R1 

PLOS ONE

Dear Dr. Zaccaria, 

I'm pleased to inform you that your manuscript has been deemed suitable for publication in PLOS ONE. Congratulations! Your manuscript is now being handed over to our production team.

Kind regards, 

on behalf of

Dr Satabdi Mitra 

Academic Editor

PLOS ONE